# TabPATE: Differentially Private Tabular In-Context Learning Without Public Data

**Dariush Wahdany** [1]   **Matthew Jagielski** [2]   **Jesse C. Cresswell** [3]   **Adam Dziedzic** [1]   **Franziska Boenisch** [1]

## Abstract

Tabular foundation models enable accurate in-context learning (ICL) from small labeled datasets, but the private records placed in context can leak through model predictions. We first show that even basic membership inference attacks succeed against tabular ICL, motivating formal privacy protection. We then introduce Tab-PATE, a differentially private PATE-style defense for tabular ICL that does not require public in-distribution data. TabPATE partitions the private context across teacher models, privately aggregates their labels on synthetic tabular queries, and releases the resulting labeled queries as a student context. Because tabular features are bounded and relatively low-dimensional, useful queries can be generated from feature ranges alone or from lightly privatized marginals. Across tabular benchmarks, TabPATE preserves competitive utility while reducing membership inference to near-random success, providing a practical path to private tabular ICL without public data.

## 1. Introduction

Tabular foundation models (TFMs) enable accurate predictions on structured data via in-context learning (ICL): a small labeled dataset is placed in the model context and used to predict labels for new queries, without task-specific training (Hollmann et al., 2023; 2025). This makes TFMs appealing in domains such as healthcare and finance, where tabular data is common and labeled data is often scarce (Tran & Byeon, 2024; Kostrzewa et al., 2025). However, the context records may contain private information.

We first verify that this concern is not merely hypothetical. In ICL, membership inference asks whether a target record was included in the private demonstration context. Adapting standard membership inference attacks (MIAs) (Shokri et al., 2017; Carlini et al., 2022a) to tabular ICL, we find that prediction behavior reveals non-trivial information about context membership. Even a passive grey-box adversary obtains $9.9\%$ true positive rate (TPR) at $1\%$ false positive rate (FPR), while stronger active attacks can reach substantially higher leakage. These results motivate mechanisms that protect the in-context data itself, rather than only the pretraining data of the foundation model.

Differential privacy (DP) (Dwork et al., 2006) provides a formal way to limit the influence of any individual private record. Yet standard DP training methods are not directly suited to tabular ICL. Differentially private stochastic gradient descent (DPSGD) (Abadi et al., 2016) requires gradients and parameter updates, whereas TFMs are used as frozen inference-time models. Private Aggregation of Teacher Ensembles (PATE) (Papernot et al., 2017; 2018), in contrast, is naturally compatible with ICL as private data can be split across teacher contexts, whose predictions are aggregated with calibrated noise. This idea underlies Prompt-PATE (Duan et al., 2023) in the language domain. Like all prior PATE-style methods, PromptPATE requires unlabeled public data from the target distribution for knowledge transfer. In sensitive tabular domains, such public in-distribution data is often unavailable.

We introduce **TabPATE**, a central-DP framework for tabular ICL that removes this public-data requirement. TabPATE partitions the private context across teacher TFMs, queries them on synthetic tabular records, privately aggregates their votes using Confident-GNMax algorithm (Papernot et al., 2018), and releases the resulting labeled synthetic records as a student context for privacy-preserving deployment. The key observation is that tabular domains differ from images and text: feature spaces are bounded, typed, and comparatively low-dimensional. Thus, useful teacher queries can be generated *without public examples*, either from a data-independent bounds-based prior or from a small DP release of marginal statistics. This yields a PATE-style defense for tabular ICL with no public in-distribution data.

We evaluate our TabPATE on standard tabular benchmarks and compare against non-private ICL, PromptPATE, query-

[1]CISPA Helmholtz Center for Information Security [2]Anthropic [3]Layer 6 AI, Toronto, Canada. Correspondence to: Dariush Wahdany <dariush.wahdany@cispa.de>.

*Proceedings of the $2^{nd}$ ICML Workshop on Foundation Models for Structured Data*, Seoul, South Korea. 2026. Copyright 2026 by the author(s).

time aggregation, and DP synthetic-data baselines. Tab-PATE preserves competitive utility at moderate privacy budgets while reducing MIA success close to random guessing. Our results suggest that public-data-free PATE-style knowledge transfer is a practical route to private deployment of TFMs. In summary, we make the following contributions:

1. We adapt membership inference to tabular ICL and show that private context records leak through model predictions, motivating formal protection.

2. We propose TabPATE, a central-DP PATE-style framework for tabular ICL that replaces public in-distribution transfer data with synthetic tabular queries.

3. We evaluate TabPATE across tabular benchmarks, showing competitive utility at moderate privacy budgets and near-random membership inference success.

## 2. Related Work

**Tabular foundation models.** TFMs perform prediction on structured data via transformer-based in-context learning (ICL) (Vaswani et al., 2017; Brown et al., 2020). Given labeled demonstrations in the context, TFMs predict labels for new queries without updating model parameters. TabPFN (Hollmann et al., 2023; 2025) is a prominent example, pretrained on synthetic tasks and competitive with strong baselines. Recent models such as TabDPT (Ma et al., 2025) and TabICL (Qu et al., 2025; 2026) further improve tabular ICL performance, with benchmarks tracking rapid progress (Erickson et al., 2025), and applications to adjacent fields (Balazadeh et al., 2025; Stith et al., 2026). We study the privacy of this inference-time learning paradigm, where sensitive records are placed directly in the model context.

**Membership inference for ICL.** MIAs (Shokri et al., 2017) are the standard empirical tool for measuring whether individual records influence model behavior. Modern attacks include calibrated likelihood-ratio attacks such as LiRA (Carlini et al., 2022a), label-only perturbation attacks (Choquette-Choo et al., 2021), and poisoning-based amplifications such as Truth Serum (Tramer et al., 2022). For text ICL, Brainwash-style attacks show that context examples can leak through resistance to adversarial relabeling (Wen et al., 2024). We adapt these attack ideas to tabular ICL to motivate the need for formal protection.

**Differential privacy and PATE.** DP (Dwork et al., 2006) bounds the effect of any single private record on a released computation. DPSGD (Abadi et al., 2016) achieves this during training via gradient clipping and noise, but is not directly applicable to frozen ICL models. PATE (Papernot et al., 2017; 2018) instead partitions private data among teachers and transfers knowledge through noisy vote aggregation, making it a natural fit for ICL. PromptPATE (Duan

et al., 2023) applies this idea to language-model prompting, but like standard PATE, relies on public in-distribution data for teacher queries. Other DP-ICL methods either answer each query privately at inference time (Wu et al., 2024) or target autoregressive text generation (Tang et al., 2024). Tab-PATE follows the PATE-style student-release approach, but removes the public-data requirement by generating synthetic tabular queries.

**DP tabular synthetic data.** DP synthetic data methods such as AIM (McKenna et al., 2022) and MST (McKenna et al., 2021) privately estimate low-dimensional marginals and post-process them with Private-PGM (Mckenna et al., 2019) to generate synthetic tables. We use such methods as baselines, but TabPATE differs in goal: it does not aim to release a synthetic version of the private dataset, but only to generate useful unlabeled queries for private PATE-style knowledge transfer.

## 3. Membership Inference in Tabular ICL

Before introducing TabPATE, we verify that private in-context records can leak through the predictions of a TFM. In standard training-based MIAs, membership asks whether a record was used for training (Shokri et al., 2017). In tabular ICL, the pretrained model is fixed; MIAs instead ask whether a target record was included in the private demonstration context used at inference time.

We consider adversaries along two axes: *output access*, where the model returns either labels only (black-box) or prediction probabilities (grey-box), and *context manipulation*, where the adversary is either passive or can inject additional examples into the context. This yields four threat models, detailed in Appendix A.1. We adapt representative attacks to each setting: label-only perturbation (Choquette-Choo et al., 2021), LiRA (Carlini et al., 2022a), Brainwash-style context manipulation (Wen et al., 2024), and LiRA with Truth Serum-style poisoning (Tramer et al., 2022).

All attacks use the same LiRA-style calibration. For each target $x_q$, we sample shadow demonstration sets, half including and half excluding $x_q$, and extract an attack-specific signal $s(x_q)$ from the model output. We fit per-sample Gaussian IN and OUT distributions and score membership by the log-likelihood ratio

$$\text{score}(x_q) = \log \frac{p_{\text{IN}}(s(x_q))}{p_{\text{OUT}}(s(x_q))}. \quad (1)$$

The attacks differ only in the signal: prediction stability under perturbations, logit-scaled confidence, number of mislabeled context copies required to flip the prediction, or confidence after poisoning.

Table 1 shows that tabular ICL leaks membership information across all threat models. Even passive grey-box access

*Table 1.* **MIA success by threat model.** Mean over five datasets with 128 shadow demonstration sets. Attack success increases with adversary capability.

| Attack | AUC | TPR@1%FPR |
|---|---|---|
| Perturbation (black-box, passive) | 0.55 | 1.4% |
| LiRA (grey-box, passive) | 0.73 | 9.9% |
| Brainwash (black-box, active) | 0.81 | 29.7% |
| LiRA + Poisoning (grey-box, active) | 0.96 | 75.4% |

yields a clear signal (0.73 AUC, 9.9% TPR at 1% FPR). Active context manipulation substantially amplifies leakage: LiRA with poisoning reaches 0.96 AUC and 75.4% TPR at 1% FPR on average. More results are presented in Appendix A. These results establish that using sensitive records directly as demonstrations creates measurable privacy risk, motivating the DP defense we introduce next.

# 4. Our TabPATE: Private Tabular ICL

TabPATE is a PATE-style mechanism for releasing a private ICL context for a TFM. Given a sensitive labeled dataset $\mathcal{D}$, our goal is to produce a new labeled student context $\widetilde{\mathcal{D}}$ that can be used for downstream predictions without further access to $\mathcal{D}$. TabPATE follows the teacher–student structure of PromptPATE (Duan et al., 2023), but removes its requirement for unlabeled public in-distribution data.

We first partition $\mathcal{D}$ into $K$ disjoint subsets. Each subset defines an ICL teacher: the same frozen TFM prompted with that subset as its demonstration context. We then query all teachers on unlabeled synthetic tabular records $\tilde{x}$ and aggregate their votes with Confident-GNMax (Papernot et al., 2018). If the teachers agree sufficiently, the mechanism releases a noisy plurality label $\tilde{y}$; otherwise it abstains. The resulting pairs $(\tilde{x}, \tilde{y})$ form the student context $\widetilde{\mathcal{D}}$, which is released and used with the frozen TFM. Since downstream predictions depend only on this released context, they incur no additional privacy cost.

The main departure from PromptPATE is how the unlabeled teacher queries are obtained. Standard PATE-style methods assume public samples from the target distribution. TabPATE instead exploits the structured nature of tabular data: features have known types and bounded ranges, so candidate records can be generated from a schema rather than a public dataset. We use a parameter $\alpha \in [0, 1]$ to split the privacy budget $\varepsilon$. When $\alpha = 0$, queries are sampled from a fixed, data-independent prior over the feature ranges, incurring no privacy cost before PATE aggregation. When $\alpha > 0$, TabPATE spends $\alpha\varepsilon$ to release simple DP marginal statistics of the private context, and samples queries from the resulting product distribution; the remaining $(1 - \alpha)\varepsilon$ is used for Confident-GNMax labeling. We empirically observe that in balanced classification setups, $\alpha = 0$ is sufficient, but

that more complex problems, such as classification of imbalanced datasets or regression (see Section 5.4) benefit from setting a small positive $\alpha > 0$. The resulting method provides central $(\varepsilon, \delta)$-DP for the released student context. The full algorithm and privacy proof are given in Appendix B.

# 5. Experiments

We evaluate whether TabPATE provides useful private ICL without public in-distribution data. We focus on four questions: (i) how much utility TabPATE preserves under DP, (ii) how it compares to other DP baselines, (iii) whether it reduces the MIA risks from §3, and (iv) whether it is able to extend from pure classification to regression setups, another core application of TFMs.

## 5.1. Setup

For the main experiments, we evaluate on five OpenML tabular classification datasets used in prior tabular ICL work: German Credit (**Credit-G**) ($n$=1000), **Blood Transfusion** ($n$=748), **Diabetes** ($n$=768), **Phoneme** ($n$=5404), and **Wilt** ($n$=4839). We use TabPFN as the TFM (Hollmann et al., 2023; 2025). Unless stated otherwise, we use $\alpha = 0$ and report the results as averaged over five random seeds. We report balanced accuracy as the main utility metric, and evaluate privacy at $\varepsilon \in \{1, 10\}$ with $\delta = 10^{-5}$.

**Baselines.** We compare TabPATE against five DP baselines covering complementary privacy strategies. As a non-ICL reference, we train a task-specific RealMLP (Holzmüller et al., 2024) with **DP-SGD**. For releasable ICL contexts without public data, **DP-Synthetic** generates a private synthetic dataset from the full sensitive data using AIM (McKenna et al., 2022) or MST (McKenna et al., 2021), while **DP-TabICL** (Carey et al., 2024) releases private prototypes; we evaluate its central-DP variant since the local-DP variant scales exponentially with the number of binary features. We also include **PromptPATE** (Duan et al., 2023), which is closest to TabPATE but assumes access to unlabeled public in-distribution data for the PATE knowledge transfer. Finally, **Query-Time** is a sample-and-aggregate baseline inspired by prior private query answering (Nissim et al., 2007; Wu et al., 2024): it partitions the private context among teachers and answers each downstream query through noisy ensemble aggregation, incurring additional privacy cost per query. A full overview of the baselines is provided in Appendix C.

## 5.2. Utility

Table 2 reports balanced accuracy averaged across the five datasets. At $\varepsilon = 10$, TabPATE achieves the best utility among the evaluated private methods, reaching 71.2% balanced accuracy without requiring public data, compared to

*Table 2.* **Balanced accuracy at different privacy budgets** (mean ± std across 5 datasets, 5 seeds).

| Approach | $\varepsilon = 1$ | $\varepsilon = 10$ |
|---|---|---|
| Non-private | $76.7 \pm 2.4\%$ | |
| DP-SGD | $51.9 \pm 0.9\%$ | $52.3 \pm 1.0\%$ |
| DP-TabICL | $51.9 \pm 0.9\%$ | $52.3 \pm 1.0\%$ |
| DP-Synthetic | $51.9 \pm 0.9\%$ | $52.3 \pm 1.0\%$ |
| Query-Time | $52.6 \pm 1.5\%$ | $61.3 \pm 4.3\%$ |
| PromptPATE[†] | $61.3 \pm 2.6\%$ | $68.6 \pm 3.0\%$ |
| **TabPATE** | $55.6 \pm 4.9\%$ | $71.2 \pm 4.6\%$ |

[†] Requires public data from target distribution; we use 30% held-out training data.

*Table 3.* **Regression results at $\varepsilon = 10$.** We report NRMSE normalized by the ground-truth standard deviation; lower is better.

| Dataset | Non-private | PromptPATE[†] | TabPATE | |
|---|---|---|---|---|
| | | | ($\alpha=0$) | ($\alpha > 0^*$) |
| Abalone | $0.645\pm0.014$ | $0.735\pm0.017$ | $0.817\pm0.033$ | $0.793\pm0.010$ |
| Calif. Housing | $0.358\pm0.010$ | $0.569\pm0.000$ | $0.741\pm0.103$ | $0.596\pm0.007$ |
| Diabetes (reg.) | $0.690\pm0.039$ | $0.792\pm0.019$ | $0.977\pm0.077$ | $0.898\pm0.005$ |
| Wine Quality | $0.695\pm0.007$ | $0.881\pm0.002$ | $0.966\pm0.058$ | $0.915\pm0.035$ |
| Bike Sharing | $0.218\pm0.005$ | $0.531\pm0.040$ | $0.617\pm0.019$ | $0.601\pm0.008$ |
| CPU Small | $0.139\pm0.004$ | $0.265\pm0.028$ | $0.462\pm0.119$ | $0.443\pm0.086$ |
| Ames Housing | $0.236\pm0.029$ | $0.562\pm0.050$ | $0.953\pm0.125$ | $0.954\pm0.110$ |
| Superconduct | $0.257\pm0.002$ | $0.504\pm0.009$ | $0.918\pm0.037$ | $0.873\pm0.058$ |
| Mean | $0.405$ | $0.605$ | $0.806$ | $0.759$ |

[†] Requires public data from the target distribution.
[*] All datasets use $\alpha = 0.05$, except Wine Quality ($0.38$) and Calif. Housing ($0.21$).

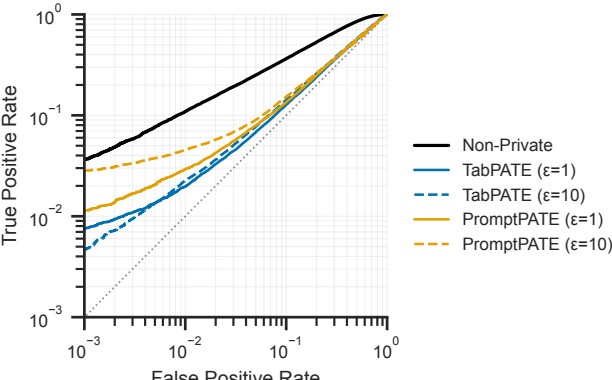

*Figure 1.* **MIA on non-private TabPFN, TabPATE, and Prompt-PATE.** Both DP methods reduce LiRA success to near-random, especially in the low-FPR region, results obtained on Credit-G.

68.6% for PromptPATE, which uses held-out in-distribution data, and 61.3% for Query-Time. At the stricter budget $\varepsilon = 1$, PromptPATE benefits from real in-distribution queries, but TabPATE still outperforms the public-data-free baselines. Full standard-accuracy results and per-dataset breakdowns are given in Appendix D.1.

### 5.3. Attack Mitigation

We next verify that our TabPATE mitigates the membership leakage observed for non-private tabular ICL. We attack the released student with the strongest passive grey-box attack, LiRA, and compare the resulting ROC curves in Figure 1. Both TabPATE and PromptPATE substantially reduce attack success in the low-FPR regime: for the non-private TabPFN, LiRA reaches an area under the curve (AUC) of about 0.76 and a TPR@1%FPR of 10.9%, whereas Tab-PATE reduces the attack to near-random performance at both $\varepsilon = 1$ and $\varepsilon = 10$. PromptPATE shows a similar mitigation effect, but requires held-out in-distribution public data. Thus, TabPATE provides comparable empirical protection against MIAs while retaining the main practical advantage of requiring no public data. We present more results obtained on larger datasets in Appendix D.3.

### 5.4. Regression

TFMs are also used for regression, so we finally evaluate whether TabPATE extends beyond classification. For regression, we replace the noisy teacher vote with a bounded noisy aggregation of teacher predictions. We first normalize the per-teacher regression target to $[0, 1]$ and clip each teacher prediction to this range to bound sensitivity. For a synthetic query $\tilde{x}$, the $K$ teachers output clipped scalar predictions $\hat{y}_1, \ldots, \hat{y}_K \in [0, 1]$, which we aggregate by their mean and perturb with Gaussian noise. Since changing one private record can affect at most one teacher, the sensitivity of the averaged prediction is bounded by $1/K$. The noisy label is then clipped to $[0, 1]$, mapped back to the original target scale, and used as a demonstration for the student. As before, releasing the student context is post-processing of the private aggregation.

Table 3 reports normalized RMSE (NRMSE; lower is better) on eight regression datasets. TabPATE remains effective without public data, though regression is more challenging than classification. PromptPATE obtains the best private utility because it uses real in-distribution public queries. Among public-data-free TabPATE settings, allocating a small budget share to marginal query generation improves over purely data-independent queries on almost all datasets, reducing mean NRMSE from $0.806$ to $0.759$. This supports the role of $\alpha > 0$ in settings where random queries insufficiently cover useful regions of the feature space.

### 6. Conclusion

We showed that tabular ICL leaks sensitive context records through membership inference and introduced TabPATE, a PATE-style DP defense for tabular foundation models. TabPATE releases a private student context using synthetic queries from feature bounds or optional DP marginals, avoiding the need for public in-distribution data. Empirically, it preserves utility while reducing MIA success to near-random levels, suggesting a practical route to private tabular ICL.

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

# A. Additional Membership Inference Details

## A.1. Threat Models and Attack Signals

We evaluate four MIAs against tabular ICL, corresponding to the two output-access levels and two context-manipulation levels in Table 4. All attacks use the LiRA-style calibration from Equation (1).

*Table 4.* **MIA methods for each threat model.**

|              | Passive      | Active            |
| ------------ | ------------ | ----------------- |
| **Black-box** | Perturbation | Brainwash         |
| **Grey-box**  | LiRA         | LiRA + Poisoning  |

**Perturbation.** For each target $x_q$, we add Gaussian noise scaled by feature standard deviation and measure the fraction of perturbed samples that preserve the original prediction:

$$s(x_q) = \frac{1}{N} \sum_{i=1}^{N} \mathbf{1}[f(x_q + \epsilon_i) = f(x_q)].$$

Members tend to be more stable because their input-output mapping is present in the context.

**LiRA.** For grey-box passive access, we use the logit-scaled confidence on the true label,

$$s(x_q, y_q) = \log \frac{p_f(y_q \mid x_q)}{1 - p_f(y_q \mid x_q)}.$$

Context members tend to receive higher confidence.

**Brainwash.** For black-box active access, the adversary repeatedly adds copies of $x_q$ with an incorrect label and records the minimum number of copies required to flip the prediction. Members require more copies because the correct label is already reinforced in the context.

**LiRA + Poisoning.** For grey-box active access, the adversary injects $k$ mislabeled copies of $x_q$ and then applies LiRA-style calibration to the poisoned signal. Poisoning separates member and non-member distributions because non-members have only the injected incorrect label in the context, while members also have the original correct record.

## A.2. Per-Dataset Perturbation Results

*Table 5.* **Label-only perturbation attack.** Results use LiRA calibration with 128 shadow demonstration sets.

| Dataset           | AUC   | TPR@1% | TPR@5% |
| ----------------- | ----- | ------ | ------ |
| German Credit     | 0.559 | 1.7%   | 6.0%   |
| Blood Transfusion | 0.509 | 1.0%   | 4.7%   |
| Diabetes          | 0.533 | 1.5%   | 5.6%   |
| Vehicle           | 0.635 | 1.3%   | 7.2%   |
| KC1               | 0.518 | 1.5%   | 5.5%   |
| Mean              | 0.551 | 1.4%   | 5.8%   |

Even under label-only passive access, the attack performs above random on all datasets, although the signal is modest (Table 5). Vehicle is the most vulnerable in this setting, consistent with the broader trend that higher-dimensional and multi-class datasets provide richer prediction signals.

## A.3. Poisoning Amplifies Membership Inference

Poisoning strongly amplifies leakage. In Table 6 with $k = 20$ mislabeled copies, the average AUC increases from 0.73 to 0.96, and several datasets become almost perfectly distinguishable. This corresponds to a small fraction of the context in our experiments, showing that active manipulation can expose membership even when passive attacks are weaker.

*Table 6.* **LiRA with demonstration poisoning.** AUC / TPR@1%FPR for different poison budgets $k$.

| Dataset | $k = 0$ | $k = 5$ | $k = 10$ | $k = 20$ |
|---|---|---|---|---|
| German Credit | 0.66 / 4.7% | 0.96 / 61.4% | 0.99 / 85.5% | 1.00 / 99.8% |
| Blood Transfusion | 0.60 / 1.1% | 0.64 / 0.2% | 0.71 / 0.0% | 0.83 / 15.0% |
| Diabetes | 0.68 / 6.6% | 0.79 / 9.1% | 0.97 / 55.5% | 1.00 / 98.2% |
| Vehicle | 0.98 / 64.3% | 1.00 / 99.7% | 1.00 / 100.0% | 1.00 / 96.8% |
| KC1 | 0.75 / 10.0% | 0.83 / 12.7% | 0.92 / 39.2% | 0.96 / 67.0% |
| Average | 0.73 / 17.3% | 0.84 / 36.6% | 0.92 / 56.0% | 0.96 / 75.4% |

## A.4. Comparison to Threshold-Based Attacks

We compare LiRA calibration to common threshold-based attacks using loss, confidence, prediction gap, entropy, correctness, and modified entropy in Table 7. LiRA is consistently strongest, with mean AUC 0.73 compared to 0.55 for the strongest threshold baseline. This supports using calibrated per-sample likelihood ratios as the main MIA evaluation.

*Table 7.* **LiRA compared to threshold-based attacks.** Mean results over five datasets.

| Attack | AUC | TPR@1%FPR |
|---|---|---|
| LiRA | 0.727 | 9.9% |
| LOSS | 0.554 | 2.6% |
| Prediction Gap | 0.554 | 2.5% |
| Confidence | 0.537 | 2.6% |
| Entropy | 0.537 | 2.6% |
| Correctness | 0.534 | 1.1% |
| Modified Entropy | 0.474 | 0.4% |

## A.5. Perturbation and Attack-Cost Ablations

The label-only perturbation attack is robust to its main hyperparameters. Varying the number of perturbations (Table 8) and perturbation scale (Table 9) changes AUC only marginally, remaining in the range 0.52–0.55 on German Credit. Thus, its limited success is due to the restrictive black-box passive setting rather than a poorly tuned perturbation configuration.

*Table 8.* **Perturbation-count ablation** on German Credit.

| Perturbations $N$ | AUC | TPR@1% |
|---|---|---|
| 10 | 0.526 | 2.1% |
| 25 | 0.531 | 2.3% |
| 50 | 0.532 | 1.1% |
| 100 | 0.524 | 1.1% |

*Table 9.* **Perturbation-scale ablation** on German Credit with $N = 50$.

| Scale $\sigma$ | AUC | TPR@1% |
|---|---|---|
| 0.05 | 0.530 | 1.7% |
| 0.10 | 0.532 | 1.1% |
| 0.20 | 0.536 | 1.3% |
| 0.50 | 0.546 | 1.8% |

Because all attacks use the same number of shadow demonstration sets, their total cost is dominated by the number of target queries. LiRA and LiRA+Poisoning require one query per target after calibration, perturbation requires $N$ queries, and Brainwash requires up to the maximum poison budget.

## A.6. Comparison to Text-Based ICL

We also compare TabPFN to text-based tabular ICL using TabLLM-style serialization (Hegselmann et al., 2023) in Table 10. Both models leak membership, although TabPFN is more vulnerable in our experiment. This suggests that membership leakage is not an artifact of one implementation, but a broader risk of using sensitive records as in-context demonstrations.

## A.7. Feature Dimensionality

We observe a moderate positive correlation between feature count and MIA success in Figure 2. Higher-dimensional datasets, such as Vehicle, tend to be more vulnerable, likely because model outputs encode richer per-record information.

*Table 10.* **Brainwash attack on TabPFN vs. TabLLM** on German Credit.

| Model | Pretraining | Brainwash AUC |
|---|---|---|
| TabPFN 2.5 | Synthetic SCMs | 0.88 |
| TabLLM (Gemini) | Web text | 0.74 |
| TabLLM (OLMo) | Web text | 0.73 |

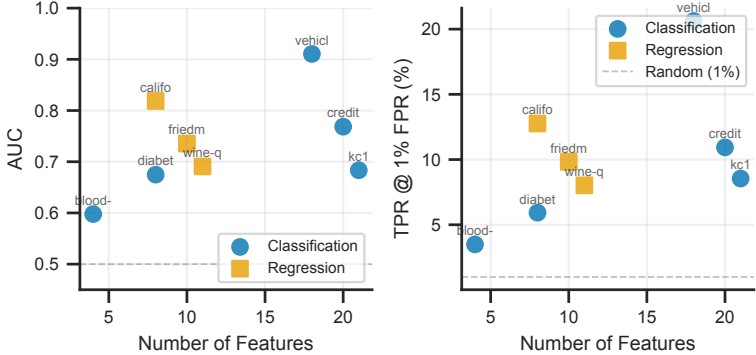

*Figure 2.* Number of features vs attack AUC across datasets. Higher-dimensional datasets exhibit increased vulnerability.

However, dimensionality alone does not fully explain leakage, which also depends on label type, dataset size, and local sample uniqueness.

## B. TabPATE Algorithm and Privacy Analysis

**Query generation.** For $\alpha = 0$, query generation is data-independent; after normalizing features to known ranges, we sample from a fixed prior such as a uniform distribution over the feature domain or a Gaussian centered at the midpoint of each range. This step has zero privacy cost. For $\alpha > 0$, we estimate per-feature means and variances with DP. Concretely, after clipping each feature vector to $\ell_2$ norm at most $C$, we release noisy versions of

$$q_1(\mathcal{D}) = |\mathcal{D}|, \qquad q_2(\mathcal{D}) = \sum_{x \in \mathcal{D}} x, \qquad q_3(\mathcal{D}) = \sum_{x \in \mathcal{D}} x \odot x.$$

The sensitivities under add/remove adjacency are $\Delta_1 = 1$, $\Delta_2 \leq C$, and $\Delta_3 \leq C^2$. We then compute $\tilde{\mu}$ and $\tilde{\sigma}$ from these noisy sufficient statistics. Sampling queries from these estimates is post-processing.

**Privacy guarantee.** TabPATE operates in the central-DP model. We state the guarantee for the released student context $\widetilde{\mathcal{D}}$.

**Theorem B.1.** *For any $\alpha \in [0, 1]$, TabPATE satisfies $(\varepsilon, \delta)$-DP under add/remove adjacency, assuming the optional marginal release is calibrated to $(\alpha\varepsilon, \delta/2)$-DP and Confident-GNMax labeling is run with budget $((1 - \alpha)\varepsilon, \delta/2)$.*

*Proof.* TabPATE is the sequential composition of two mechanisms. The first mechanism, present only when $\alpha > 0$, releases noisy marginal statistics. Gaussian noise is calibrated to the sensitivities of the count, sum, and sum-of-squares queries so that their joint release satisfies $(\alpha\varepsilon, \delta/2)$-DP. The derived means, variances, and synthetic query samples are post-processing and incur no additional privacy loss. When $\alpha = 0$, no private statistic is released, so this step is $(0, 0)$-DP.

The second mechanism labels synthetic queries using Confident-GNMax (Papernot et al., 2018). The teachers are defined on disjoint partitions of $\mathcal{D}$, so changing one record affects at most one teacher vote. Confident-GNMax adds calibrated noise to the vote aggregation and tracks privacy expenditure across answered queries; we stop labeling once the allocated budget

$$((1 - \alpha)\varepsilon, \delta/2)$$

is exhausted.

---

**Algorithm 1** TabPATE for private tabular ICL. Setting $\alpha = 0$ skips the marginal-release step and samples queries from a fixed bounds-based prior.

---

**Require:** Private data $\mathcal{D}$, number of teachers $K$, target budget $(\varepsilon, \delta)$, budget split $\alpha$, query budget $M$, feature bounds.
**Ensure:** Released student context $\widetilde{\mathcal{D}}$.
1: Partition $\mathcal{D}$ into disjoint subsets $\mathcal{D}_1, \ldots, \mathcal{D}_K$.
2: Define teacher $T_k(\cdot)$ as the frozen TFM prompted with $\mathcal{D}_k$.
3: **if** $\alpha > 0$ **then**
4:     Clip features and release noisy count, sum, and sum-of-squares under budget $(\alpha\varepsilon, \delta/2)$.
5:     Derive $\tilde{\mu}$ and $\tilde{\sigma}$ by post-processing the noisy aggregates.
6: **end if**
7: Initialize $\widetilde{\mathcal{D}} \leftarrow \emptyset$.
8: **for** $j = 1, \ldots, M$ **do**
9:     **if** $\alpha = 0$ **then**
10:         Sample $\tilde{x}_j$ from a fixed prior over feature bounds.
11:     **else**
12:         Sample $\tilde{x}_j \sim \mathcal{N}(\tilde{\mu}, \text{diag}(\tilde{\sigma}^2))$, clipped to feature bounds.
13:     **end if**
14:     Query teachers and form vote counts $n_c = \sum_{k=1}^{K} \mathbf{1}[T_k(\tilde{x}_j) = c]$.
15:     Apply Confident-GNMax with budget $((1-\alpha)\varepsilon, \delta/2)$.
16:     **if** Confident-GNMax returns label $\tilde{y}_j$ **then**
17:         $\widetilde{\mathcal{D}} \leftarrow \widetilde{\mathcal{D}} \cup \{(\tilde{x}_j, \tilde{y}_j)\}$.
18:     **end if**
19: **end for**
20: **RETURN** $\widetilde{\mathcal{D}}$.

---

By sequential composition, the combined mechanism satisfies

$$(\alpha\varepsilon, \delta/2) + ((1-\alpha)\varepsilon, \delta/2) = (\varepsilon, \delta).$$

Finally, releasing and using the student context $\widetilde{\mathcal{D}}$ is post-processing of the private aggregation outputs, so downstream predictions with the frozen TFM do not incur additional privacy cost. $\qquad\square$

## C. Baselines

In the following, we present our baselines in more detail. We also provide an overview in Table 11.

**DP-SGD (non-ICL baseline).** As an additional reference point, we include a non-ICL baseline. We train a RealMLP (Holzmüller et al., 2024) with DP-SGD.

**DP-Synthetic Data.** DP-Synthetic generates DP synthetic data from the full private dataset via AIM (McKenna et al., 2022) or MST (McKenna et al., 2021) and uses them as demonstrations for ICL of a student model.

**DP-TabICL.** DP-TabICL (Carey et al., 2024) releases $k$ private prototypes from the private data. It splits the data into stratified splits. They propose an LDP and GDP variant. The former is limited to less than 14 binary features due to exponential scaling in the number of dimensions, hence we evaluate the latter, which computes DP per-feature means with the Laplace mechanism.

**PromptPATE.** PromptPATE (Duan et al., 2023) partitions the private data among ICL teachers and uses public in-distribution data for the private knowledge transfer via Confident-GNMax (Papernot et al., 2018). It requires access to unlabeled public data from the same distribution as the private data, which limits applicability in sensitive domains.

**Query-Time.** Query-Time, inspired by (Nissim et al., 2007; Wu et al., 2024), partitions the context data among teachers and directly answers queries at inference time with aggregated noisy ensemble predictions. Each query incurs additional privacy cost, requiring the system to stop once the budget is exhausted.

*Table 11.* **Overview of DP defenses.**

| Defense | No Public Data | Releasable | Privacy Cost |
|---|:---:|:---:|:---:|
| DP-SGD (RealMLP)[‡] | ✓ | ✓ | Training |
| DP-Synthetic | ✓ | ✓ | Marginals |
| DP-TabICL | ✓ | ✓ | Data release |
| Query-Time | ✓ | ✗ | Aggregation (per-query) |
| PromptPATE | ✗ | ✓ | Aggregation |
| **TabPATE (ours)** | ✓ | ✓ | Marginals & Aggregation |

[‡] Non-ICL baseline: trains a task-specific model with DP-SGD instead of using ICL.

*Table 12.* **Accuracy at different privacy budgets** (mean $\pm$ std across 5 datasets, 5 seeds).

| | Balanced Acc | | Standard Acc | |
|---|:---:|:---:|:---:|:---:|
| Approach | $\varepsilon = 1$ | $\varepsilon = 10$ | $\varepsilon = 1$ | $\varepsilon = 10$ |
| Non-private | $76.7 \pm 2.4\%$ | | $83.0 \pm 2.1\%$ | |
| DP-SGD | $51.9 \pm 0.9\%$ | $52.3 \pm 1.0\%$ | $73.1 \pm 2.1\%$ | $73.2 \pm 2.1\%$ |
| DP-TabICL | $51.9 \pm 0.9\%$ | $52.3 \pm 1.0\%$ | $73.1 \pm 2.1\%$ | $73.2 \pm 2.1\%$ |
| DP-Synthetic | $51.9 \pm 0.9\%$ | $52.3 \pm 1.0\%$ | $73.1 \pm 2.1\%$ | $73.2 \pm 2.1\%$ |
| Query-Time | $52.6 \pm 1.5\%$ | $61.3 \pm 4.3\%$ | $56.3 \pm 2.7\%$ | $71.2 \pm 3.4\%$ |
| PromptPATE[†] | $61.3 \pm 2.6\%$ | $68.6 \pm 3.0\%$ | $73.6 \pm 1.9\%$ | $78.6 \pm 2.2\%$ |
| **TabPATE** | $55.6 \pm 4.9\%$ | $71.2 \pm 4.6\%$ | $77.2 \pm 5.6\%$ | $78.3 \pm 2.7\%$ |

[†] Requires public data from target distribution; we use 30% held-out training data.

# D. Additional Experimental Results

## D.1. Extended Utility Results

This section reports the full utility results omitted from the main paper for space. Table 12 includes both balanced and standard accuracy averaged across datasets, while Table 13 provides the per-dataset balanced accuracy.

## D.2. Effect of Query Generation

Table 14 illustrates why marginal-based queries are useful in some settings. On Wilt, a highly imbalanced dataset, random uniform or Gaussian queries provide poor balanced accuracy, while DP marginal queries place more queries near the relevant data region and recover high utility.

## D.3. Larger Classification Datasets

We evaluate TabPATE on larger datasets to test whether public-data-free PATE-style transfer remains useful beyond the small OpenML benchmarks (Bischl et al., 2021). Table 15 reports utility at $\varepsilon = 10$. We also measure passive grey-box MIA vulnerability of non-private tabular ICL on these datasets in Table 16. Non-private ICL remains vulnerable even at tens of thousands of samples, although leakage decreases on some larger datasets, consistent with the privacy onion effect (Carlini et al., 2022b) with potential fairness implications (Cresswell, 2025).

*Table 13.* **Balanced accuracy at $\varepsilon = 1$ and $\varepsilon = 10$ for each dataset** (mean $\pm$ std across seeds). TabPATE requires no public data; PromptPATE uses 30% held-out in-distribution data.

| | Blood Transfusion | | Credit-G | | Diabetes | | Phoneme | | Wilt | |
|---|---|---|---|---|---|---|---|---|---|---|
| Approach | $\varepsilon = 1$ | $\varepsilon = 10$ | $\varepsilon = 1$ | $\varepsilon = 10$ | $\varepsilon = 1$ | $\varepsilon = 10$ | $\varepsilon = 1$ | $\varepsilon = 10$ | $\varepsilon = 1$ | $\varepsilon = 10$ |
| Non-private | 65.0$\pm$4.9 | | 67.2$\pm$3.2 | | 72.7$\pm$4.2 | | 88.0$\pm$1.6 | | 89.4$\pm$6.2 | |
| DP-Synthetic | 50.3$\pm$1.3 | 50.3$\pm$0.9 | 50.0$\pm$0.0 | 50.0$\pm$0.0 | 50.0$\pm$0.0 | 50.0$\pm$0.0 | 58.1$\pm$10.2 | 60.2$\pm$11.1 | 50.0$\pm$0.0 | 50.0$\pm$0.0 |
| Query-Time | 53.8$\pm$3.0 | 56.0$\pm$3.0 | 50.7$\pm$1.0 | 52.8$\pm$0.9 | 56.3$\pm$3.6 | 63.1$\pm$0.5 | 51.0$\pm$0.8 | 62.6$\pm$6.2 | 51.1$\pm$1.1 | 72.1$\pm$11.5 |
| PromptPATE[†] | 62.7$\pm$9.2 | 60.0$\pm$5.8 | 50.9$\pm$1.1 | 54.6$\pm$5.3 | 56.3$\pm$7.1 | 63.6$\pm$4.6 | 68.5$\pm$4.6 | 72.3$\pm$3.2 | 67.8$\pm$17.0 | 89.7$\pm$5.0 |
| TabPATE ($\alpha{=}0$) | 55.1$\pm$4.9 | 55.0$\pm$5.1 | 54.3$\pm$1.4 | 56.5$\pm$2.2 | 73.5$\pm$2.2 | 75.7$\pm$1.2 | 70.3$\pm$2.0 | 72.5$\pm$1.3 | 56.8$\pm$7.9 | 62.7$\pm$5.7 |
| TabPATE | 53.5$\pm$0.7 | 51.9$\pm$0.0 | 50.0$\pm$0.0 | 50.0$\pm$0.0 | 61.6$\pm$0.0 | 63.8$\pm$0.4 | 66.1$\pm$3.2 | 67.4$\pm$4.3 | 59.9$\pm$14.0 | 88.4$\pm$5.1 |

[†]Requires public data from target distribution; we use 30% held-out training data.

*Table 14.* **Query generation on Wilt** at $\varepsilon = 10$.

| Query strategy | Balanced accuracy |
|---|---|
| DP Marginal | 0.88 $\pm$ 0.05 |
| Gaussian prior | 0.63 $\pm$ 0.06 |
| Uniform prior | 0.62 $\pm$ 0.13 |

*Table 15.* **Utility on larger classification datasets** at $\varepsilon = 10$ (mean $\pm$ std). PromptPATE requires public in-distribution data.

| Dataset | Non-private | PromptPATE[†] | TabPATE-R | TabPATE-M |
|---|---|---|---|---|
| Electricity | .858$\pm$.001 | .758$\pm$.004 | .738$\pm$.003 | .731$\pm$.006 |
| Bank Marketing | .745$\pm$.001 | .619$\pm$.025 | .638$\pm$.033 | .585$\pm$.116 |
| Adult Income | .781$\pm$.002 | .734$\pm$.009 | .775$\pm$.030 | .781$\pm$.024 |
| Madelon | .907$\pm$.008 | .539$\pm$.056 | .529$\pm$.057 | .520$\pm$.023 |

[†]Uses public in-distribution data for teacher queries.

*Table 16.* **Non-private MIA leakage on larger datasets.** Passive grey-box LiRA attack.

| Dataset | $n$ | AUC | TPR@1%FPR |
|---|---|---|---|
| Electricity | 38,474 | .711 | .104 |
| Bank Marketing | 45,211 | .632 | .086 |
| Adult Income | 48,842 | .576 | .038 |
| Madelon | 2,600 | .915 | .330 |

