# OpenReview forum: "TabPATE: Differentially Private Tabular In-Context Learning Without Public Data"
_ICML.cc/2026/Workshop/FMSD — FMSD @ ICML 2026 Poster_

### Official Review · Reviewer_kFif · 2026-05-21
**Promising public-data-free privacy protection for Tabular ICL, but query generation and evaluation need stronger validation**

**Rating:** 6
**Confidence:** 2

**Review:**

**Summary**

This paper proposes a differentially private PATE-style framework for tabular foundation models used in in-context learning. The authors first show that private records placed in the context can leak through membership inference attacks, especially under active attack settings. To address this, the proposed method, TabPATE, partitions private data into teacher contexts, queries them using synthetic tabular records, privately aggregates teacher labels, and releases a labeled synthetic student context for downstream prediction. A key contribution is that TabPATE does not require public in-distribution data, instead using feature bounds or differentially private marginal statistics to generate queries. Empirically, the method shows promising utility-privacy tradeoffs for classification and reduces membership inference risk, although its performance depends on query generation quality and the evaluation could be expanded.

**Strengths**

1. The paper addresses an important problem. It identifies that privacy risks can arise not only from the pretraining of tabular foundation models, but also from inference-time context data.

2. The authors evaluate multiple threat models, including passive and active settings, and show that tabular in-context learning can leak membership information.

3. The proposed method is conceptually clean and well aligned with the PATE framework.

4. The use of synthetic tabular queries from feature bounds or differentially private marginals is practically appealing, especially when public in-distribution data are unavailable.

**Areas for Improvement**

1. The experimental evaluation could be strengthened. The main classification experiments use only five OpenML datasets, many of which are relatively small.

2. The baseline results need more explanation. In the main classification table, DP-SGD, DP-TabICL, and DP-Synthetic have nearly identical balanced accuracy values at both $\epsilon = 1$ and $\epsilon = 10$. This raises questions about whether these baselines were fully tuned or whether they collapsed to near-random balanced accuracy.

3. The query generation strategy could be analyzed more deeply. The paper argues that tabular feature spaces are bounded and relatively low-dimensional, but synthetic queries from simple priors may still fall outside realistic high-density regions, especially for correlated, mixed-type, or imbalanced datasets. The appendix result on Wilt shows that differentially private marginal queries perform much better than Gaussian or uniform priors, suggesting that query generation quality can strongly affect performance.

4. The regression results are weaker than the classification results. PromptPATE achieves better regression performance, while TabPATE with $\alpha > 0$ still has substantially higher mean NRMSE. The paper should more clearly discuss this limitation and identify when public-data-free query generation is insufficient.

5. The paper motivates the problem using several attacks, but the defense evaluation mainly focuses on passive grey-box LiRA. Evaluating active attacks against the released student context would make the privacy claims more convincing.

**Detailed Comments**

1. Please provide more details on the privacy accounting for Confident-GNMax, including the noise scale, threshold, number of teachers, query budget, and how many synthetic labels are typically released before the privacy budget is exhausted.

2. It would be helpful to clarify how the number of teachers $K$ and the number of synthetic queries $M$ are selected.

3. It would be helpful to explain why DP-SGD, DP-TabICL, and DP-Synthetic have nearly identical performance in the main classification table.

4. The paper would benefit from ablations on $\alpha$, $K$, $M$, query generation strategy, and student context size.

5. The method assumes bounded and typed tabular features. It would be helpful to discuss practical handling of categorical variables, high-cardinality categorical features, missing values, and strong feature correlations.

---

### Official Review · Reviewer_EyZK · 2026-05-21

**Rating:** 7
**Confidence:** 3

**Review:**

# Summary

The paper investigates privacy leakage risks of in-context records in tabular foundation models (TFMs) and proposes TabPATE, a central-DP framework based on PromptPATE. TabPATE generates synthetic tabular queries and privately aggregates teacher predictions to construct student context without requiring public in-distribution data. Experimental results show that TabPATE achieves effective MIA mitigation and better utility than existing public-data-free baselines, with only slight utility degradation compared to PromptPATE.

# Strength

1. The paper provides a systematic evaluation of membership inference attacks against tabular ICL under diverse threat models, showing that context records can leak substantial membership information even under passive grey-box adversaries.
2. TabPATE extends PromptPATE to the tabular ICL setting without requiring public in-distribution data. The query generation mechanism allocates privacy budget ($\alpha \epsilon$) to generate synthetic tabular queries from DP marginal statistics, which is practical and well motivated for tabular domains.
3. Extensive experiments demonstrate that TabPATE effectively mitigates membership leakage while maintaining competitive utility compared to existing public-data-free baselines.

# Weakness

1. Although the paper evaluates multiple membership inference threat models, the privacy mitigation experiments mainly focus on the passive grey-box LiRA attack. Additional analysis under stronger active attack settings could further improve the completeness of the empirical study.
2. The effectiveness of TabPATE appears sensitive to the privacy budget allocation parameter $\alpha$, which controls the trade-off between synthetic query quality and private aggregation noise. While larger $\alpha$ values may help synthetic queries better approximate the true data distribution, the utility improvements brought by DP marginals appear somewhat dataset-dependent in Tables 3, 13, and 15. A more systematic analysis of this trade-off would strengthen the paper.

---

### Official Review · Reviewer_8vmy · 2026-05-22
**Overall, a novel paper -- clear accept**

**Rating:** 8
**Confidence:** 4

**Review:**

Summary:
This paper addresses the privacy risks of tabular foundation models, where sensitive records are placed directly in the model's context for in-context learning. The authors first demonstrate that membership inference attacks succeed against tabular ICL across multiple threat models. They then propose TabPATE, a PATE-style differentially private defense that partitions the private context among teacher models, queries them on synthetic tabular records, privately aggregates their votes via Confident-GNMax, and releases the resulting labeled synthetic records as a student context.

Strengths:
1. The paper provides a convincing empirical demonstration that tabular ICL leaks membership information.
2. The main contribution, replacing public in-distribution transfer data with synthetic tabular queries, is well-motivated for sensitive domains where public in-distribution data is genuinely unavailable.
3. TabPATE is conceptually elegant: the α parameter provides a smooth interpolation between data-independent and marginal-informed query generation, and the overall framework is easy to implement and reason about. The simplicity is a strength for practical adoption.
4. The paper compares against five DP baselines covering different privacy strategies, with a clear summary table.

Areas for Improvement:
1. There are 2 other related works that could be discussed or cited here, PATE-GAN and G-PATE
2. In Table 2, DP-SGD, DP-TabICL, and DP-Synthetic have the identical results?
3. The main evaluation uses only five small OpenML datasets. The authors could have more evaluation datasets.
4. The authors could provide more details about the experiments, like the number of teachers, query budget, and clipping norm.

Detailed Comments:
1. The authors could experiment on and explain how sensitive are results to the number of teachers? What is the tradeoff between more teachers and fewer demonstrations per teacher.

Justification of Score:
The paper addresses a valid problem and proposes a clean and practical solution. However, the authors could increase the evaluation scale and provide more details about the experiments.